# BLURRING DIFFUSION MODELS

**Emiel Hoogeboom**
Google Research, Brain Team,
Amsterdam, Netherlands

**Tim Salimans**
Google Research, Brain Team,
Amsterdam, Netherlands

## ABSTRACT

Recently, Rissanen et al. (2022) have presented a new type of diffusion process for generative modeling based on *heat dissipation*, or *blurring*, as an alternative to isotropic Gaussian diffusion. Here, we show that blurring can equivalently be defined through a Gaussian diffusion process with non-isotropic noise. In making this connection, we bridge the gap between inverse heat dissipation and denoising diffusion, and we shed light on the inductive bias that results from this modeling choice. Finally, we propose a generalized class of diffusion models that offers the best of both standard Gaussian denoising diffusion and inverse heat dissipation, which we call *Blurring Diffusion Models*.

## 1 INTRODUCTION

Diffusion models are becoming increasingly successful for image generation, audio synthesis and video generation. Diffusion models define a (stochastic) process that destroys a signal such as an image. In general, this process adds Gaussian noise to each dimension independently. However, data such as images clearly exhibit multi-scale properties which such a diffusion process ignores.

Recently, the community is looking at new destruction processes which are referred to as deterministic or 'cold' diffusion (Rissanen et al., 2022; Bansal et al., 2022). In these works, the diffusion process is either deterministic or close to deterministic. For example, in (Rissanen et al., 2022) a diffusion model that incorporates heat dissipation is proposed, which can be seen as a form of blurring. Blurring is a natural destruction for images, because it retains low frequencies over higher frequencies.

However, there still exists a considerable gap between the visual quality of standard denoising diffusion models and these new deterministic diffusion models. This difference cannot be explained away by a limited computational budget: A standard diffusion model can be trained with relative little compute (about one to four GPUs) with high visual quality on a task such as unconditional CIFAR10 generation[1]. In contrast, the visual quality of *deterministic* diffusion models have been

---

[1]An example of a denoising diffusion implementation https://github.com/w86763777/pytorch-ddpm

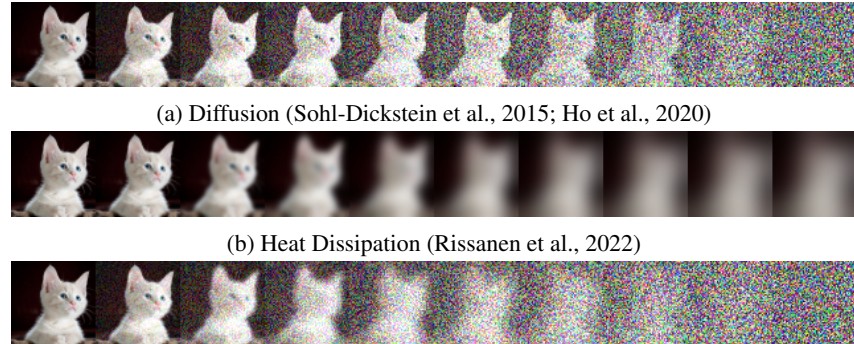

(a) Diffusion (Sohl-Dickstein et al., 2015; Ho et al., 2020)

(b) Heat Dissipation (Rissanen et al., 2022)

(c) Blurring Diffusion

Figure 1: Comparison between standard diffusion, heat dissipation and blurring diffusion.

much worse so far. In addition, fundamental questions remain around the justification of deterministic diffusion models: Does their specification offer any guarantees about being able to model the data distribution?

In this work, we aim to resolve the gap in quality between models using blurring and additive noise. We present Blurring Diffusion Models, which combine blurring (or heat dissipation) and additive Gaussian noise. We show that the given process can have Markov transitions and that the denoising process can be written with diagonal covariance in frequency space. As a result, we can use modern techniques from denoising diffusion. Our model generates samples with higher visual quality, which is evidenced by better FID scores.

## 2 BACKGROUND

### 2.1 DIFFUSION MODELS

Diffusion models (Sohl-Dickstein et al., 2015; Song & Ermon, 2019; Ho et al., 2020) learn to generate data by denoising a pre-defined destruction process which is named the diffusion process. Commonly, the diffusion process starts with a datapoint and gradually adds Gaussian noise to the datapoint. Before defining the generative process, this diffusion process needs to be defined. Following the definition of (Kingma et al., 2021) the diffusion process can be written as:

$$q(\boldsymbol{z}_t|\boldsymbol{x}) = \mathcal{N}(\boldsymbol{z}_t|\alpha_t \boldsymbol{x}, \sigma_t^2 \mathbf{I}), \tag{1}$$

where $\boldsymbol{x}$ represents the data and $\boldsymbol{z}_t$ are the noisy latent variables. Since $\alpha_t$ is monotonically decreasing and $\sigma_t$ is monotonically increasing, the information from $\boldsymbol{x}$ in $\boldsymbol{z}_t$ will be gradually destroyed as $t$ increases. Assuming that the above process defined by Equation 1 is Markov, it has transition distributions for $\boldsymbol{z}_t$ given $\boldsymbol{z}_s$ where $0 \leq s < t$:

$$q(\boldsymbol{z}_t|\boldsymbol{z}_s) = \mathcal{N}(\boldsymbol{z}_t|\alpha_{t|s}\boldsymbol{z}_s, \sigma_{t|s}^2 \mathbf{I}), \tag{2}$$

where $\alpha_{t|s} = \alpha_t/\alpha_s$ and $\sigma_{t|s}^2 = \sigma_t^2 - \alpha_{t|s}^2 \sigma_s^2$. A convenient property is that the grid of timesteps can be defined arbitrarily and does not depend on the specific spacing of $s$ and $t$. We let $T = 1$ denote the last diffusion step where $q(\boldsymbol{z}_T|\boldsymbol{x}) \approx \mathcal{N}(\boldsymbol{z}_T|\mathbf{0}, \mathbf{I})$, a standard normal distribution. Unless otherwise specified, a time step lies in the unit interval $[0, 1]$.

**The Denoising Process** Another important distribution is the true denoising distribution $q(\boldsymbol{z}_s|\boldsymbol{z}_t, \boldsymbol{x})$ given a datapoint $\boldsymbol{x}$. Using that $q(\boldsymbol{z}_s|\boldsymbol{z}_t, \boldsymbol{x}) \propto q(\boldsymbol{z}_t|\boldsymbol{z}_s)q(\boldsymbol{z}_s|\boldsymbol{x})$ one can derive that:

$$q(\boldsymbol{z}_s|\boldsymbol{z}_t, \boldsymbol{x}) = \mathcal{N}(\boldsymbol{z}_s|\boldsymbol{\mu}_{t \to s}, \sigma_{t \to s}^2 \mathbf{I}), \tag{3}$$

where

$$\sigma_{t \to s}^2 = \left(\frac{1}{\sigma_s^2} + \frac{\alpha_{t|s}^2}{\sigma_{t|s}^2}\right)^{-1} \text{ and } \boldsymbol{\mu}_{t \to s} = \sigma_{t \to s}^2 \left(\frac{\alpha_{t|s}}{\sigma_{t|s}^2}\boldsymbol{z}_t + \frac{\alpha_s}{\sigma_s^2}\boldsymbol{x}\right) \tag{4}$$

To generate data, the true denoising process is approximated by a learned denoising process $p(\boldsymbol{z}_s|\boldsymbol{z}_t)$, where the datapoint $\boldsymbol{x}$ is replaced by a prediction from a learned model. The model distribution is then given by

$$p(\boldsymbol{z}_s|\boldsymbol{z}_t) = q(\boldsymbol{z}_s|\boldsymbol{z}_t, \hat{\boldsymbol{x}}(\boldsymbol{z}_t)), \tag{5}$$

where $\hat{\boldsymbol{x}}(\boldsymbol{z}_t)$ is a prediction provided by a neural network. As shown by Song et al. (2020), the true $q(\boldsymbol{z}_s|\boldsymbol{z}_t) \to q(\boldsymbol{z}_s|\boldsymbol{z}_t, \boldsymbol{x} = \mathbb{E}[\boldsymbol{x}|\boldsymbol{z}_t])$ as $s \to t$, which justifies this choice of model: If the generative model takes sufficiently small steps, and if $\hat{\boldsymbol{x}}(\boldsymbol{z}_t)$ is sufficiently expressive, the model can learn the data distribution exactly.

Instead of directly predicting $\boldsymbol{x}$, diffusion models can also model $\hat{\boldsymbol{\epsilon}}_t = f_\theta(\boldsymbol{z}_t, t)$, where $f_\theta$ is a neural net, so that:

$$\hat{\boldsymbol{x}} = \boldsymbol{z}_t/\alpha_t - \sigma_t/\alpha_t \hat{\boldsymbol{\epsilon}}_t, \tag{6}$$

which is inspired by the reparametrization to sample from Equation 1 which is $\boldsymbol{z}_t = \alpha_t \boldsymbol{x} + \sigma_t \boldsymbol{\epsilon}_t$. This parametrization is called the epsilon parametrization and empirically leads to better sample quality than predicting $\boldsymbol{x}$ directly (Ho et al., 2020).

**Optimization** As shown in (Kingma et al., 2021), a continuous-time variational lower bound on the model log likelihood $\log p(x)$ is given by the following expectation over squared reconstruction errors:

$$\mathcal{L} = \mathbb{E}_{t \sim \mathcal{U}(0,1)} \mathbb{E}_{\boldsymbol{\epsilon}_t \sim \mathcal{N}(0,\mathbf{I})} [w(t) || f_\theta(\boldsymbol{z}_t, t) - \boldsymbol{\epsilon}_t ||^2], \tag{7}$$

where $\boldsymbol{z}_t = \alpha_t \boldsymbol{x}_t + \sigma_t \boldsymbol{\epsilon}_t$. When these terms are weighted appropriately with a particular weight $w(t)$, this objective corresponds to a variational lowerbound on the model likelihood $\log p(x)$. However, empirically a constant weighting $w(t) = 1$ has been found to be superior for sample quality.

## 2.2 Inverse Heat Dissipation

Instead of adding increasing amounts of Gaussian noise, Inverse Heat Dissipation Models (IHDMs) use heat dissipation to destroy information (Rissanen et al., 2022). They observe that the Laplace partial differential equation for heat dissipation

$$\frac{\partial}{\partial t} \boldsymbol{z}(i, j, t) = \Delta \boldsymbol{z}(i, j, t) \tag{8}$$

can be solved by a diagonal matrix in the frequency domain of the cosine transform if the signal is discretized to a grid. Letting $\boldsymbol{z}_t$ denote the solution to the Laplace equation at time-step $t$, this can be efficiently computed by:

$$\boldsymbol{z}_t = \mathbf{A}_t \boldsymbol{z}_0 = \mathbf{V} \mathbf{D}_t \mathbf{V}^{\mathrm{T}} \boldsymbol{z}_0 \tag{9}$$

where $\mathbf{V}^{\mathrm{T}}$ denotes a Discrete Cosine Transform (DCT) and $\mathbf{V}$ denotes the Inverse DCT and $\boldsymbol{z}_0, \boldsymbol{z}_t$ should be considered vectorized over spatial dimensions to allow for matrix multiplication. The diagonal matrix $\mathbf{D}_t$ is the exponent of a weighting matrix for frequencies $\boldsymbol{\Lambda}$ and the dissipation time $t$ so that $\mathbf{D}_t = \exp(-\boldsymbol{\Lambda} t)$. For the specific definition of $\boldsymbol{\Lambda}$ see Appendix A. In (Rissanen et al., 2022) marginal distribution of the diffusion process is defined as:

$$q(\boldsymbol{z}_t | \boldsymbol{x}) = \mathcal{N}(\boldsymbol{z}_t | \mathbf{A}_t \boldsymbol{x}, \sigma^2 \mathbf{I}). \tag{10}$$

The intermediate diffusion state $\boldsymbol{z}_t$ is thus constructed by adding a fixed amount of noise to an increasingly blurred data point, rather than adding an increasing amount of noise as in the DDPMs described in Section 2.1. The generative process in (Rissanen et al., 2022) approximately inverts the heat dissipation process with a learned generative model:

$$p(\boldsymbol{z}_{t-1} | \boldsymbol{z}_t) = \mathcal{N}(\boldsymbol{z}_{t-1} | f_\theta(\boldsymbol{z}_t), \delta^2 \mathbf{I}), \tag{11}$$

where the mean for $\boldsymbol{z}_{t-1}$ is directly learned with a neural network $f_\theta$ and has fixed scalar variance $\delta^2$. Similar to DDPMs, the IHDM model is learned by sampling from the forward process $\boldsymbol{z}_t \sim q(\boldsymbol{z}_t | \boldsymbol{x})$ for a random timestep $t$, and then minimizing the squared reconstruction error between the model $f_\theta(\boldsymbol{z}_t)$ and a ground truth target, which in this case is given by $\mathbb{E}(\boldsymbol{z}_{t-1} | \boldsymbol{x}) = \mathbf{A}_{t-1} \boldsymbol{x}$, yielding the training loss $\mathcal{L} = \mathbb{E}_{t \sim \mathcal{U}(1,...,T)} \mathbb{E}_{\boldsymbol{z}_t \sim q(\boldsymbol{z}_t | \boldsymbol{x})} \left[ || \mathbf{A}_{t-1} \boldsymbol{x} - f_\theta(\boldsymbol{z}_t, t) ||^2 \right]$.

**Arbitrary Dissipation Schedule** There is no reason why the conceptual time-steps of the model should match perfectly with the dissipation time. Therefore, in (Rissanen et al., 2022) $\mathbf{D}_t = \exp(-\boldsymbol{\Lambda} \tau_t)$ is redefined where $\tau_t$ monotonically increases with $t$. The variable $\tau_t$ has a very similar function as $\alpha_t$ and $\sigma_t$ in noise diffusion: it allows for arbitrary dissipation schedules with respect to the conceptual time-steps $t$ of the model.

To avoid confusion, note that in (Rissanen et al., 2022) $k$ is used as the conceptual time for the diffusion process, $t_k$ is the dissipation time and $\boldsymbol{u}_k$ denotes the latent variables. In this paper, $t$ is the conceptual time and $\boldsymbol{z}_t$ denotes the latent variables in pixel space. Then $\tau_t$ is used to denote dissipation time.

**Open Questions** Certain questions remain: (1) Can the heat dissipation process be Markov and if so what is $q(\boldsymbol{z}_t | \boldsymbol{z}_s)$? (2) Is the true inverse heating process also isotropic, as the generative process in Equation 11? (3) Finally, are there alternatives to predicting the mean of the previous time-step?

In the following section it will turn out that: (1) Yes, the process can be Markov. As a result, denoising equations similar to the ones for standard diffusion can be derived. (2) No, the generative process is not isotropic, although it is diagonal in the frequency domain. As a consequence, the

correct amount of noise (per-dimension) can be derived analytically instead of choosing it heuristically. This also guarantees that the model $p(z_s|z_t)$ can actually express the true $q(z_s|z_t)$ as $s \to t$, because it is known to tend towards $q(z_s|z_t, x = \mathbb{E}[x|z_t])$ (Song et al., 2020). (3) Yes, processes like heat dissipation can be parametrized similar to the epsilon parametrization in standard diffusion models.

## 3 HEAT DISSIPATION AS GAUSSIAN DIFFUSION

Here we reinterpret the heat dissipation process as a form of Gaussian diffusion similar to that used in (Ho et al., 2020; Sohl-Dickstein et al., 2015; Song & Ermon, 2019) and others. Throughout this paper, multiplication and division between two vectors is defined to be elementwise. We start with the definition of the marginal distribution from (Rissanen et al., 2022):

$$q(z_t|x) = \mathcal{N}(z_t|\mathbf{A}_t x, \sigma^2 \mathbf{I}) \tag{12}$$

where $\mathbf{A}_t = \mathbf{V}\mathbf{D}_t\mathbf{V}^{\mathrm{T}}$ denotes the blurring or dissipation operation as defined in the previous section. Throughout this section we let $\mathbf{V}^{\mathrm{T}}$ denote the orthogonal DCT, which is a specific normalization setting of the DCT. Under the change of variables $u_t = \mathbf{V}^{\mathrm{T}} z_t$ we can write the diffusion process in frequency space for $u_t$:

$$q(u_t|u_x) = \mathcal{N}(u_t|d_t \cdot u_x, \sigma^2 \mathbf{I}) \tag{13}$$

where $u_x = \mathbf{V}^{\mathrm{T}} x$ is the frequency response of $x$, $d_t$ is the diagonal of $\mathbf{D}_t$ and vector multiplication is done elementwise. Whereas we defined $\mathbf{D}_t = \exp(-\mathbf{\Lambda}\tau_t)$ we let $\lambda$ denote the diagonal of $\mathbf{\Lambda}$ so that $d_t = \exp(-\lambda\tau_t)$. Essentially, $d_t$ multiplies higher frequencies with smaller values.

Equation 13 shows that the marginal distribution of the frequencies $u_t$ is fully factorized over its scalar elements $u_t^{(i)}$ for each dimension $i$. Similarly, the inverse heat dissipation model $p_\theta(u_s|u_t)$ is also fully factorized. We can thus equivalently describe the heat dissipation process (and its inverse) in scalar form for each dimension $i$:

$$q(u_t^{(i)}|u_0^{(i)}) = \mathcal{N}(u_t^{(i)}|d_t^{(i)}u_0^{(i)}, \sigma^2) \quad \Leftrightarrow \quad u_t^{(i)} = d_t^{(i)}u_0^{(i)} + \sigma\epsilon_t, \text{ with } \epsilon_t \sim \mathcal{N}(0,1). \tag{14}$$

This equation can be recognized as a special case of the standard Gaussian diffusion process introduced in Section 2.1. Let $s_t$ denote a standard diffusion process in frequency space, so $s_t^{(i)} = \alpha_t u_0^{(i)} + \sigma_t \epsilon_t$. We can see that Rissanen et al. (2022) have chosen $\alpha_t = d_t^{(i)}$ and $\sigma_t^2 = \sigma^2$. As shown by Kingma et al. (2021), from a probabilistic perspective only the ratio $\alpha_t/\sigma_t$ matters here, not the particular choice of the individual $\alpha_t, \sigma_t$. This is true because all values can simply be re-scaled without changing the distributions in a meaningful way.

This means that, rather than performing blurring and adding fixed noise, the heat dissipation process can be equivalently defined as a relatively standard Gaussian diffusion process, albeit in frequency space. The non-standard aspect here is that the diffusion process in (Rissanen et al., 2022) is defined in the frequency space $u$, and that it uses a separate noise schedule $\alpha_t, \sigma_t$ for each of the scalar elements of $u$: i.e. the noise in this process is *non-isotropic*. That the marginal variance $\sigma^2$ is shared between all scalars $u^{(i)}$ under their specification does not reduce its generality: the ratio $\alpha_t/\sigma$ can be freely determined per dimension, and this is all that matters.

**Markov transition distributions** An open question in the formulation of heat dissipation models by Rissanen et al. (2022) was whether or not there exists a Markov process $q(u_t|u_s)$ that corresponds to their chosen marginal distribution $q(z_t|x)$. Through its equivalence to Gaussian diffusion shown above, we can now answer this question affirmatively. Using the results summarized in Section 2.1, we have that this process is given by

$$q(u_t|u_s) = \mathcal{N}(u|\alpha_{t|s}u_s, \sigma_{t|s}^2) \tag{15}$$

where $\alpha_{t|s} = \alpha_t/\alpha_s$ and $\sigma_{t|s}^2 = \sigma_t^2 - \alpha_{t|s}^2 \sigma_s^2$. Substituting in the choices of Rissanen et al. (2022), $\alpha_t = d_t$ and $\sigma_t^{(i)} = \sigma$, then gives

$$\alpha_{t|s} = d_t/d_s \quad \text{and} \quad \sigma_{t|s}^2 = (1 - (d_t/d_s)^2)\sigma^2. \tag{16}$$

Note that if $d_t$ is chosen so that it contains lower values for higher frequencies, then $\sigma_{t|s}$ will add *more noise* on the *higher frequencies* per timestep. The heat dissipation model thus destroys information more quickly for those frequencies as compared to standard diffusion.

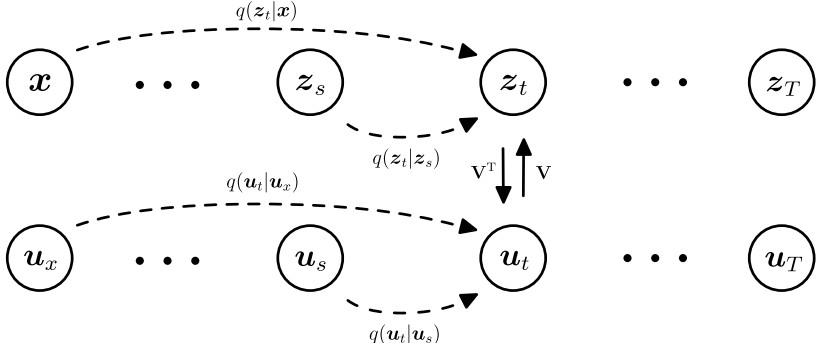

Figure 2: A blurring diffusion process with latent variable $z_0, \dots, z_1$ is diagonal (meaning can be factorized over dimensions) in frequency space, under the change of variable $u_t = \mathbf{V}^T z_t$. This results in a corresponding diffusion process in frequency space $u_0, \dots, u_1$.

**Denoising Process**  Using again the results from Section 2.1, we can find an analytic expression for the inverse heat dissipation process:

$$q(u_s | u_t, x) = \mathcal{N}(u_s | \mu_{t \to s}, \sigma^2_{t \to s}),  \tag{17}$$

where

$$\sigma^2_{t \to s} = \left( \frac{1}{\sigma^2_s} + \frac{\alpha^2_{t|s}}{\sigma^2_{t|s}} \right)^{-1} \text{and } \mu_{t \to s} = \sigma^2_{t \to s} \left( \frac{\alpha_{t|s}}{\sigma^2_{t|s}} u_t + \frac{\alpha_s}{\sigma^2_s} u_x \right).  \tag{18}$$

Except for $u_x$, we can again plug in the expressions derived above in terms of $d_t, \sigma^2$. The analysis in Section 2.1 then allows predicting $\epsilon_t$ using a neural network to complete the model, as is done in standard denoising diffusion models. In comparison (Rissanen et al., 2022) predict $\mu_{t \to s}$ directly, which is theoretically equally general but has been found to lead to inferior sample quality. Furthermore, they instead chose to use a single scalar value for $\sigma^2_{t \to s}$ for all time-steps: the downside of this is that it loses the guarantee of correctness as $s \to t$ as described in Section 2.1.

## 4 BLURRING DIFFUSION MODELS

In this section we propose Blurring Diffusion Models. Using the analysis from Section 3, we can define this model in frequency space as a Gaussian diffusion model, with different schedules for the dimensions. Blurring diffusion places more on emphasis low frequencies which are visually more important, and it may also avoid over-fitting to high frequencies. It is important how the model is parametrized and what the specific schedules for $\alpha_t$ and $\sigma_t$ are. Different from traditional models, the diffusion process is defined in a frequency space:

$$q(u_t | u_x) = \mathcal{N}(u_t | \alpha_t u_x, \sigma^2_t \mathbf{I})  \tag{19}$$

and different frequencies may diffuse at a different rate, which is controlled by the values in the vectors $\alpha_t, \sigma_t$ (although we will end up picking the same scalar value for all dimensions in $\sigma_t$). Recall that the denoising distribution is then given by $q(u_s | u_t, x) = \mathcal{N}(u_s | \mu_{t \to s}, \sigma^2_{t \to s})$ as specified in Equation 17.

**Learning and Parametrization**  An important reason for the performance of modern diffusion models is the parametrization. Learning $\mu_{t \to s}$ directly turns out to be difficult for neural networks and instead an approximation for $x$ is learned which is plugged into the denoising distributions, often indirectly via an epsilon parametrization (Ho et al., 2020). Studying the re-parametrization of Equation 19:

$$u_t = \alpha_t u_x + \sigma_t u_{\epsilon,t} \quad \text{where} \quad u_x = \mathbf{V}^T x \text{ and } u_{\epsilon,t} = \mathbf{V}^T \epsilon_t  \tag{20}$$

and take that as inspiration for the way we parametrize our model:

$$\left( u_t - \sigma_t \hat{u}_{\epsilon,t} \right)/\alpha_t = \hat{u}_x,  \tag{21}$$

---

**Algorithm 1** Generating Samples

Sample $\boldsymbol{z}_T \sim \mathcal{N}(0, \mathbf{I})$
**for** $t$ in $\{\frac{T}{T}, \ldots, \frac{1}{T}\}$ where $s = t - 1/T$ **do**
$\quad \boldsymbol{u}_t = \mathbf{V}^\mathrm{T} \boldsymbol{z}$ and $\hat{\boldsymbol{u}}_{\epsilon,t} = \mathbf{V}^\mathrm{T} f_\theta(\boldsymbol{z}, t)$
$\quad$ Compute $\boldsymbol{\sigma}_{t \to s}$ and $\hat{\boldsymbol{\mu}}_{t \to s}$ with Eq. 18, 23
$\quad$ Sample $\boldsymbol{\epsilon} \sim \mathcal{N}(0, \mathbf{I})$
$\quad \boldsymbol{z} \leftarrow \mathbf{V}(\hat{\boldsymbol{\mu}}_{t \to s} + \boldsymbol{\sigma}_{t \to s} \boldsymbol{\epsilon})$

---

**Algorithm 2** Optimizing Blurring Diffusion

Sample $t \sim \mathcal{U}(0, 1)$
Sample $\boldsymbol{\epsilon} \sim \mathcal{N}(0, \mathbf{I})$
Minimize $||\boldsymbol{\epsilon} - f_\theta(\mathbf{V}\boldsymbol{\alpha}_t\mathbf{V}^\mathrm{T}\boldsymbol{x} + \sigma_t\boldsymbol{\epsilon}, t)||^2$

---

which is the blurring diffusion counterpart of Equation 6 from standard diffusion models. Although it is convenient to express our diffusion and denoising processes in frequency space, neural networks have been optimized to work well in standard pixel space. It is for this reason that the neural network $f_\theta$ takes as input $\boldsymbol{z}_t = \mathbf{V}\boldsymbol{u}_t$ and predicts $\hat{\boldsymbol{\epsilon}}_t$. After prediction we can always easily transition back and forth between frequency space if needed using the DCT matrix $\mathbf{V}^\mathrm{T}$ and inverse DCT matrix $\mathbf{V}$. This is how $\hat{\boldsymbol{u}}_{\epsilon,t} = \mathbf{V}^\mathrm{T}\hat{\boldsymbol{\epsilon}}_t$ is obtained. Using this parametrization for $\hat{\boldsymbol{x}}$ and after transforming to frequency space $\hat{\boldsymbol{u}}_x = \mathbf{V}^\mathrm{T}\hat{\boldsymbol{x}}$ we can compute $\hat{\boldsymbol{\mu}}_{t \to s}$ using Equation 18 where $\boldsymbol{u}_x$ is replaced by the prediction $\hat{\boldsymbol{u}}_x$ to give:

$$p(\boldsymbol{u}_s | \boldsymbol{u}_t) = q(\boldsymbol{u}_s | \boldsymbol{u}_t, \hat{\boldsymbol{u}}_x) = \mathcal{N}(\boldsymbol{u}_s | \hat{\boldsymbol{\mu}}_{t \to s}, \boldsymbol{\sigma}_{t \to s}) \tag{22}$$

for which $\hat{\boldsymbol{\mu}}_{t \to s}$ can be simplified further in terms of $\hat{\boldsymbol{u}}_{\epsilon,t}$ instead of $\hat{\boldsymbol{u}}_x$:

$$\hat{\boldsymbol{\mu}}_{t \to s} = \boldsymbol{\sigma}_{t \to s}^2 \left( \frac{\boldsymbol{\alpha}_{t|s}}{\boldsymbol{\sigma}_{t|s}^2} \boldsymbol{u}_t + \frac{1}{\boldsymbol{\alpha}_{t|s} \boldsymbol{\sigma}_s^2} (\boldsymbol{u}_t - \sigma_t \hat{\boldsymbol{u}}_{\epsilon,t}) \right). \tag{23}$$

**Optimization** Following the literature (Ho et al., 2020) we optimize an unweighted squared error in pixel space:

$$\mathcal{L} = \mathbb{E}_{t \sim \mathcal{U}(0,1)} \mathbb{E}_{\boldsymbol{\epsilon}_t \sim \mathcal{N}(0,\mathbf{I})}[||f_\theta(\boldsymbol{z}_t, t) - \boldsymbol{\epsilon}_t||^2], \quad \text{where } \boldsymbol{z}_t = \mathbf{V}(\boldsymbol{\alpha}_t\mathbf{V}^\mathrm{T}\boldsymbol{x}_t + \boldsymbol{\sigma}_t\mathbf{V}^\mathrm{T}\boldsymbol{\epsilon}_t). \tag{24}$$

Alternatively, one can derive a variational bound objective which corresponds to a different weighting as explained in section 2.1. However, it is known that such objectives tend to result in inferior sample quality (Ho et al., 2020; Nichol & Dhariwal, 2021).

**Noise and Blurring Schedules** To specify the blurring process precisely, the schedules for $\boldsymbol{\alpha}_t$, $\boldsymbol{\sigma}_t$ need to be defined for $t \in [0, 1]$. For $\boldsymbol{\sigma}_t$ we choose the same value for all frequencies, so it suffices to give a schedule for a scalar value $\sigma_t$. The schedules are constructed by combining a typical Gaussian noise diffusion schedule (specified by scalars $a_t, \sigma_t$) with a blurring schedule (specified by the vectors $\boldsymbol{d}_t$).

For the noise schedule, following (Nichol & Dhariwal, 2021) we choose a variance preserving cosine schedule meaning that $\sigma_t^2 = 1 - a_t^2$, where $a_t = \cos(t\pi/2)$ for $t \in [0, 1]$. To avoid instabilities when $t \to 0$ and $t \to 1$, the log signal to noise ratio ($\log a_t^2/\sigma_t^2$) is at maximum $+10$ for $t = 0$ and at least $-10$ for $t = 1$. See (Kingma et al., 2021) for more details regarding the relation between the signal to noise ratio and $a_t, \sigma_t$. For the blurring schedule, we use the relation from (Rissanen et al., 2022) that a Gaussian blur with scale $\sigma_B$ corresponds to dissipation with time $\tau = \sigma_B^2/2$. Empirically we found the blurring schedule:

$$\sigma_{B,t} = \sigma_{B,\max} \sin(t\pi/2)^2 \tag{25}$$

to work well, where $\sigma_{B,\max}$ is a tune-able hyperparameter that corresponds to the maximum blur that will be applied to the image. This schedule in turn defines the dissipation time via $\tau_t = \sigma_{B,t}^2/2$. As described in Equation 23, the denoising process divides elementwise by the term $\boldsymbol{\alpha}_{t|s} = \boldsymbol{\alpha}_t/\boldsymbol{\alpha}_s$. If one would naively use $\boldsymbol{d}_t = \exp(-\boldsymbol{\lambda}\tau_t)$ for $\boldsymbol{\alpha}_t$ and equivalently for step $s$, then the term $\boldsymbol{d}_t/\boldsymbol{d}_s$ could contain very small values for high frequencies. As a result, an undesired side-effect is that small errors may be amplified by many steps of the denoising process. Therefore, we modify the procedure slightly and let:

$$\boldsymbol{d}_t = (1 - d_{\min}) \cdot \exp(-\boldsymbol{\lambda}\tau_t) + d_{\min}, \tag{26}$$

where we set $d_{\min} = 0.001$. This blurring transformation damps frequencies to a small value $d_{\min}$ and at the same time the denoising process amplifies high frequencies less aggressively. Because

(Rissanen et al., 2022) did not use the denoising process, this modification was not necessary for their model. Combining the Gaussian noise schedule $(a_t, \sigma_t)$ with the blurring schedule $(\boldsymbol{d}_t)$ we obtain:

$$\boldsymbol{\alpha}_t = a_t \cdot \boldsymbol{d}_t \quad \text{and} \quad \boldsymbol{\sigma}_t = \mathbf{1}\sigma_t, \tag{27}$$

where $\mathbf{1}$ is a vector of ones. See Appendix A for more details on the implementation and specific settings.

### 4.1 A NOTE ON THE GENERALITY

In general, an orthogonal base $\boldsymbol{u}_x = \mathbf{V}^{\mathrm{T}}\boldsymbol{x}$ that has a diagonal diffusion process $q(\boldsymbol{u}_t|\boldsymbol{u}_x) = \mathcal{N}(\boldsymbol{u}_t|\boldsymbol{\alpha}_t\boldsymbol{u}_x, \sigma_t^2\mathbf{I})$ corresponds to the following process in pixel space:

$$q(\boldsymbol{z}_t|\boldsymbol{x}) = \mathcal{N}(\boldsymbol{z}_t|\mathbf{V}\mathrm{diag}(\boldsymbol{\alpha}_t)\mathbf{V}^{\mathrm{T}}\boldsymbol{x}, \mathbf{V}\mathrm{diag}(\boldsymbol{\sigma}_t^2)\mathbf{V}^{\mathrm{T}}) \quad \text{where} \quad \boldsymbol{u}_t = \mathbf{V}^{\mathrm{T}}\boldsymbol{z}_t, \tag{28}$$

where $\mathrm{diag}$ transforms a vector to a diagonal matrix. More generally, a diffusion process defined in any invertible basis change $\boldsymbol{u}_x = \mathbf{P}^{-1}\boldsymbol{x}$ corresponds to the following diffusion process in pixel space:

$$q(\boldsymbol{z}_t|\boldsymbol{x}) = \mathcal{N}(\boldsymbol{z}_t|\mathbf{P}\mathrm{diag}(\boldsymbol{\alpha}_t)\mathbf{P}^{-1}\boldsymbol{x}, \mathbf{P}\mathrm{diag}(\boldsymbol{\sigma}_t^2)\mathbf{P}^{\mathrm{T}}) \quad \text{where} \quad \boldsymbol{u}_t = \mathbf{P}^{-1}\boldsymbol{z}_t. \tag{29}$$

As such, this framework enables a larger class of diffusion models, with the guarantees of standard diffusion models.

## 5 RELATED WORK

Score-based diffusion models (Sohl-Dickstein et al., 2015; Song & Ermon, 2019; Ho et al., 2020) have become increasingly successfully in modelling different types of data, such as images (Dhariwal & Nichol, 2021), audio (Kong et al., 2021), and steady states of physical systems (Xu et al., 2022). Most diffusion processes are diagonal, meaning that they can be factorized over dimensions. The vast majority relies on independent additive isotropic Gaussian noise as a diffusion process.

Several diffusion models use a form of super-resolution to account for the multi-scale properties in images (Ho et al., 2022; Jing et al., 2022). These methods still rely on additive isotropic Gaussian noise, but have explicit transitions between different resolutions. In other works (Serrà et al., 2022; Kawar et al., 2022) diffusion models are used to restore predefined corruptions on image or audio data, although these models do not generate data from scratch. Theis et al. (2022) discuss non-isotropic Gaussian diffusion processes in the context of lossy compression. They find that non-isotropic Gaussian diffusion, such as our blurring diffusion models, can lead to improved results if the goal is to encode data with minimal mean-squared reconstruction loss under a reconstruction model that is constrained to obey the ground truth marginal data distribution, though the benefit over standard isotropic diffusion is greater for different objectives.

Recently, several works introduce other destruction processes as an alternative to Gaussian diffusion with little to no noise. Although pre-existing works invert fixed amounts of blur (Kupyn et al., 2018; Whang et al., 2022), in (Rissanen et al., 2022) blurring is directly built into the diffusion process via heat dissipation. Similarly, in (Bansal et al., 2022) several (possibly deterministic) destruction mechanisms are proposed which are referred to as 'cold diffusion'. However, the generative processes of these approaches may not be able to properly learn the reveres process if they do not satisfy the condition discussed in section 2.1. Furthermore in (Lee et al., 2022) a process is introduced that combines blurring and noise and is variance preserving in frequency space, which may not be the ideal inductive bias for images. Concurrently, in (Daras et al., 2022) a method is introduced that can incorporate blurring with noise, although sampling is done differently. For all these approaches, there is still a considerably gap in performance compared to standard denoising diffusion.

## 6 EXPERIMENTS

### 6.1 COMPARISON WITH DETERMINISTIC AND DENOISING DIFFUSION MODELS

In this section our proposed Blurring Diffusion Models are compared to their closest competitor in literature, IHDMs (Rissanen et al., 2022), and to Cold Diffusion Models (Bansal et al., 2022). In addition, they are also compared to a denoising diffusion baseline similar to DDPMs (Ho et al., 2020) which we refer to as Denoising Diffusion.

Table 1: Sample quality on CIFAR10 measured in FID score, lower is better.

| CIFAR10 | FID |
| --- | --- |
| Cold Diffusion (Blur)* | 80.08 |
| IHDM (Rissanen et al., 2022) | 18.96 |
| Soft Diffusion (Daras et al., 2022) | 4.64 |
| Denoising Diffusion | 3.58 |
| Blurring Diffusion (ours) | **3.17** |

* Not unconditional, starts from blurred image.

Table 2: Sample quality on LSUN churches $128 \times 128$ measured in FID score.

| Model | FID |
| --- | --- |
| IHDM (Rissanen et al., 2022) | 45.1 |
| Denoising Diffusion | 4.68 |
| Blurring Diffusion (ours) | **3.88** |

**CIFAR10** The first generation task is generating images when trained on the CIFAR10 dataset (Krizhevsky et al., 2009). For this task, we run the blurring diffusion model and the denoising diffusion baseline both using the same UNet architecture as their noise predictor $f_\theta$. Specifically, the UNet operates at resolutions $32 \times 32$, $16 \times 16$ and $8 \times 8$ with 256 channels at each level. At every resolution, the UNet has 3 residual blocks associated with the down-sampling section and another 3 blocks for the up-sampling section. Furthermore, the UNet has self-attention at resolutions $16 \times 16$ and $8 \times 8$ with a single head. Although IHDMs used only 128 channels on the $32 \times 32$ resolutions, they use 256 channels on all other resolutions, they include the $4 \times 4$ resolution and use 4 blocks instead of 3 blocks. Also see Appendix A.2.

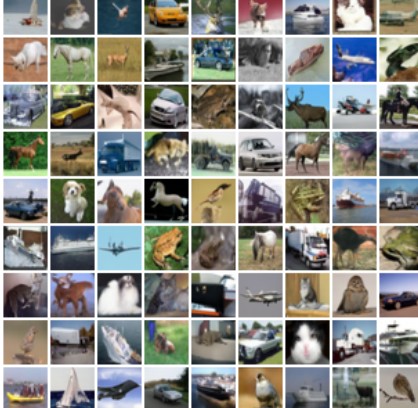

Figure 3: Samples from a Blurring Diffusion Model trained on CIFAR10.

To measure the visual quality of the generated samples we use the FID score measured on 50000 samples drawn from the models, after 2 million steps of training. As can be seen from these scores (Table 1), the blurring diffusion models are able to generate images with a considerable higher quality than IHDMs, as well as other similar approaches in literature. Our blurring diffusion models also outperform standard denoising diffusion models, although the difference in performance is less pronounced in that case. Random samples drawn from the model are depicted in Figure 3.

**LSUN Churches** Secondly, we test the performance of the model when trained on LSUN Churches with a resolution of $128 \times 128$. Again, a UNet architecture is used for the noise prediction network $f_\theta$. This time the UNet operates on 64 channels for the $128 \times 128$ resolution, 128 channels for the $64 \times 64$ resolution, 256 channels for the $32 \times 32$ resolution, 384 channels for the $16 \times 16$ resolution, and 512 channels for the $8 \times 8$ resolution. At each resolution there are two sections with 3 residual blocks, with self-attention on the resolutions $32 \times 32$, $16 \times 16$, and $8 \times 8$. The models in (Rissanen et al., 2022) use more channels at each resolution level but only 2 residual blocks (see Appendix A.2).

The visual quality is measured by computing the FID score on 10000 samples drawn from trained models. From these scores (Table 2) again we see that the blurring diffusion models generate higher quality images than IHDMs. Furthermore, Blurring Diffusion models also outperform denoising diffusion

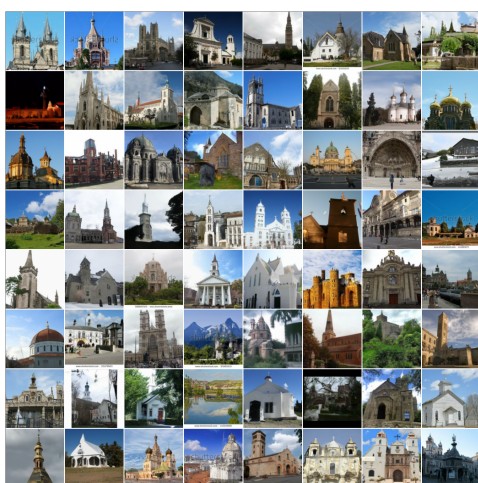

Figure 4: Samples from a Blurring Diffusion model trained on LSUN churches $128 \times 128$.

models, although again the difference in performance is smaller in that comparison. See Appendix B for more experiments.

Table 3: Blurring Diffusion Models with different maximum noise values

| $\sigma_{B,\max}$ | CIFAR10 | LSUN Churches (128×) |
|---|---|---|
| 0 | 3.60 | 4.68 |
| 1 | 3.49 | 4.42 |
| 10 | 3.26 | 3.65 |
| 20 | 3.17 | 3.88 |

Table 4: Different maximum noise levels and schedules on CIFAR10

| $\sigma_{B,\max}$ | $\sigma_{B,\max}\sin(t\pi/2)^2$ | $\sigma_{B,\max}\sin(t\pi/2)$ |
|---|---|---|
| 0 | 3.60 | 3.58 |
| 1 | 3.49 | 3.37 |
| 10 | 3.26 | 4.24 |
| 20 | 3.17 | 6.54 |

## 6.2 COMPARISON BETWEEN DIFFERENT NOISE LEVELS AND SCHEDULES

In this section we analyze the models from above, but with different settings in terms of maximum blur ($\sigma_{B,\max}$) and two different noise schedule ($\sin^2$ and $\sin$). The models where $\sigma_{B,\max} = 0$ are equivalent to a standard denoising diffusion model. For CIFAR10, the best performing model uses a blur of $\sigma_{B,\max} = 20$ which has an FID of 3.17 over 3.60 when no blur is applied, as can be seen in Table 3. The difference compared to the model with $\sigma_{B,\max} = 10$ is relatively small, with an FID of 3.26. For LSUN Churches, the the best performing model uses a little less blur $\sigma_{B,\max} = 10$ although performance is again relatively close to the model with $\sigma_{B,\max} = 20$. When comparing the $\sin^2$ schedule with a $\sin$ schedule, the visual quality measured by FID score seems to be much better for the $\sin^2$ schedule (Table 4). In fact, for higher maximum blur the $\sin^2$ schedule performs much better. Our hypothesis is that the $\sin$ schedule blurs too aggressively, whereas the graph of a $\sin^2$ adds blur more gradually at the beginning of the diffusion process near $t = 0$.

Interesting behaviour of blurring diffusion models is that models with higher maximum blur ($\sigma_{B,\max}$) converge more slowly, but when trained long enough outperform models with less blur. When comparing two blurring models with $\sigma_{B,\max}$ set to either 1 or 20, the model with $\sigma_{B,\max} = 20$ has better visual quality only after roughly 200K training steps for CIFAR10 and 1M training steps for LSUN churches. It seems that higher blur takes more time to train, but then learns to fit the data better. Note that an exception was made for the evaluation of the CIFAR10 models where $\sigma_{B,\max}$ is 0 and 1, as those models show over-fitting behaviour and have better FID at 1 million steps than at 2 million steps. Regardless of this selection advantage, they are outperformed by blurring diffusion models with higher $\sigma_{B,\max}$.

## 7 LIMITATIONS AND CONCLUSION

In this paper we introduced *blurring diffusion models*, a class of generative models generalizing over the Denoising Diffusion Probabilistic Models (DDPM) of Ho et al. (2020) and the Inverse Heat Dissipation Models (IHDM) of Rissanen et al. (2022). In doing so, we showed that blurring data, and several other such deterministic transformations with addition of fixed variance Gaussian noise, can equivalently be defined through a Gaussian diffusion process with non-isotropic noise. This allowed us to make connections to the literature on non-isotropic diffusion models (e.g. Theis et al., 2022), which allows us to better understand the inductive bias imposed by this model class. Using our proposed model class, we were able to generate images with improved perceptual quality compared to both DDPM and IHDM baselines.

A limitation of blurring diffusion models is that the use of blur has a regularizing effect: When using blur it takes longer to train a generative model to convergence. Such as regularizing effect is often beneficial, and can lead to improved sample quality as we showed in Section 6, but may not be desirable when very large quantities of training data are available. As we discuss in Section 4, the expected benefit of blurring is also dependent on our particular objective, and will differ for different ways of measuring sample quality: We briefly explored this in Section 6, but we leave a more exhaustive exploration of the tradeoffs in this model class for future work.

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

## A  ADDITIONAL DETAILS ON BLURRING DIFFUSION

In this section we provide additional details for blurring diffusion models. In particular, we provide some pseudo-code to show the essential steps that are needed to compute the variables associated with the diffusion process.

### A.1  PSEUDO-CODE OF DIFFUSION AND DENOISING PROCESS

Firstly, the procedure to compute the frequency scaling ($d_t$) is given below:

```
def get_frequency_scaling(t, min_scale=0.001):
  # compute dissipation time
  sigma_blur = sigma_blur_max * sin(t * pi / 2)^2
  dissipation_time = sigma_t^2 / 2

  # compute frequencies
  freq = pi * linspace(0, img_dim-1, img_dim) / img_dim
  labda = freqs[None, :, None, None]^2 + freqs[None, None, :, None]^2

  # compute scaling for frequencies
  scaling = exp(-labda * dissipation_time) * (1 - min_scale)
  scaling = scaling + min_scale
  return scaling
```

Note here the computation of $\Lambda$ is from (Rissanen et al., 2022) and the variable 'scaling' refers to $d_t$ in equations. Next, we can define a wrapper function to return the required $\alpha_t, \sigma_t$ values.

```
def get_alpha_sigma(t):
  freq_scaling = get_frequency_scaling(t)
  a, sigma = get_noise_scaling_cosine(t)
  alpha = a * freq_scaling   # Combine dissipation and scaling.
  return alpha, sigma
```

Which also requires a function to obtain the noise parameters. We use a typical cosine schedule for which the pseudo-code is given below:

```
def get_noise_schaling_cosine(t, logsnr_min=-10, logsnr_max=10):
  limit_max = arctan(exp(-0.5 * logsnr_max))
  limit_min = arctan(exp(-0.5 * logsnr_min)) - limit_max
  logsnr = -2 * log(tan(limit_min * t + limit_max))

  # Transform logsnr to a, sigma.
  return sqrt(sigmoid(logsnr)), sqrt(sigmoid(-logsnr))
```

To train the model we desire samples from $q(\boldsymbol{u}_t|\boldsymbol{u}_x)$. In the pseudo-code below, the inputs ($\boldsymbol{x}$) and outputs ($\boldsymbol{z}_t, \boldsymbol{\epsilon}_t$) are defined in pixel space. Recall that $\boldsymbol{z}_t = \mathbf{V}\boldsymbol{u}_t = \mathrm{IDCT}(\boldsymbol{u}_t)$ and then:

```
def diffuse(x, t):
  x_freq = DCT(x)

  alpha, sigma = get_alpha_sigma(t)
  eps = random_normal_like(x)

  # Since we chose sigma to be a scalar, eps does not need to be
  # passed through a DCT/IDCT in this case.
  z_t = IDCT(alpha * x_freq) + sigma * eps

  return z_t, eps
```

Given samples $\boldsymbol{z}_t$ from the diffusion process one can now directly define the mean squared error loss on epsilon as defined below:

```
def loss(x):
  t = random_uniform(0, 1)
  z_t, eps = diffuse(x, t)
  error = (eps - neural_net(z_t, t))^2
  return mean(error)
```

Finally, to sample from the model we repeatedly sample from $p(\boldsymbol{z}_{t-1/T}|\boldsymbol{z}_t)$ for the grid of timesteps $t = T, T - 1/T \ldots, 1/T$.

```
def denoise(z_t, t, delta=1e-8):
  alpha_s, sigma_s = get_alpha_sigma(t - 1 / T)
  alpha_t, sigma_t = get_alpha_sigma(t)

  # Compute helpful coefficients.
  alpha_ts = alpha_t / alpha_s
  alpha_st = 1 / alpha_ts
  sigma2_ts = (sigma_t^2 - alpha_ts^2 * sigma_s^2)

  # Denoising variance.
  sigma2_denoise = 1 / clip(
    1 / clip(sigma_s^2, min=delta) +
    1 / clip(sigma_t^2 / alpha_ts^2 - sigma_s^2, min=delta),
    min=delta)

  # The coefficients for u_t and u_eps.
  coeff_term1 = alpha_ts * sigma2_denoise / (sigma2_ts + delta)
  coeff_term2 = alpha_st * sigma2_ts / clip(sigma_s^2, min=delta)

  # Get neural net prediction.
  hat_eps = neural_net(z_t, t)

  # Compute terms.
  u_t = DCT(z_t)
  term1 = IDCT(coeff_term1 * u_t)
  term2 = IDCT(coeff_term2 * u_t - sigma_t * DCT(hat_eps)))
  mu_denoise = term1 + term2

  # Sample from the denoising distribution.
  eps = random_normal_like(mu_denoise)
  return mu_denoise + IDCT(sqrt(sigma2_denoise) * eps)
```

More efficient implementations that use less DCT calls are also possible when the denoising function is directly defined in frequency space. This is not really an issue however, because compared to the neural network the DCTs are relatively cheap. Additionally, several values are clipped to a minimum of $10^{-8}$ to avoid numerically unstable divisions.

In the sampling process of standard diffusion, before using the prediction $\hat{\epsilon}$ the variable is transformed to $\hat{x}$, clipped and then transformed back to $\hat{\epsilon}$. This procedure is known to improve visual quality scores for standard denoising diffusion, but it is not immediately clear how to apply the technique in the case of blurring diffusion. For future research, finding a reliable technique to perform clipping without introducing frequency artifacts may be important.

## A.2 HYPERPARAMETER SETTINGS

In the experiments, the neural network function ($f_\theta$ in equations) is implemented as a UNet architecture, as is typical in modern diffusion models (Ho et al., 2020). For the specific architecture details see Table 5. Note that as is standard in UNet architectures, there is an downsample and upsample path. Following the common notation, the hyperparameter 'ResBlocks / Stage' denotes the blocks per stage per upsample/downsample path. Thus, a level with 3 ResBlocks per stage as in total $3 + (3 + 1) = 7$ ResBlocks, where the $(3 + 1)$ originates from the upsample path which always uses an additional block. In addition, the downsample / upsample blocks also apply an additional ResBlock. All models where optimized with Adam, with a learning rate of $2 \cdot 10^{-4}$ and batch size

128 for CIFAR-10 and a learning rate of $1 \cdot 10^{-4}$ and batch size 256 for the LSUN models. All methods are evaluated with an exponential moving average computed with a decay of 0.9999.

Table 5: Architecture Settings

| Experiment | Channels | Attention Resolutions | Head dim | ResBlocks / Stage | Channel Multiplier | Dropout |
|---|---|---|---|---|---|---|
| CIFAR10 | 256 | 8, 16 | 256 | 3 | 1, 1, 1 | 0.2 |
| LSUN Churches 64 | 128 | 8, 16, 32 | 64 | 3 | 1, 2, 3, 4 | 0.2 |
| LSUN Churches 128 | 64 | 8, 16, 32 | 64 | 3 | 1, 2, 4, 6, 8 | 0.1 |

## B  ADDITIONAL EXPERIMENTS

In this section, some additional information regarding the experiments are shown. In Table 6 the FID score on the eval set of CIFAR10 and LSUN churches $128 \times 128$ is presented. The best performing models match with the results in the main text on train FID. For CIFAR10, we also report the Inception Score which corresponds to the certainty of the Inception classifier. Here the results are less clear, because all models have roughly similar scores. The best performing model uses $\sigma_{B,\max} = 10$ and achieves 9.59. To confirm that the loss and parametrization are important, the best CIFAR10 model (with $\sigma_{B,\max} = 20$) is trained using a mean squared error on $x - \hat{x}$ when predicting $\hat{x}$, but this only achieves 23.9 FID versus the 3.17 of the epsilon parametrization. This diminished performance is also observed for standard diffusion (Ho et al., 2020). Furthermore, as an ablation study we trained the best performing model in the frequency domain (where the UNet takes as input $u_t$). This model only produced gray samples with some checkerboard artifacts, and had a higher loss throughout training. This indicates that learning a UNet directly in frequency space is not straightforward.

Table 6: Blurring Diffusion Models with different maximum noise values (eval FID) and Inception Score (IS) for CIFAR10.

| $\sigma_{B,\max}$ | CIFAR10 (FID eval) | CIFAR10 (IS) | LSUN Churches (eval FID) |
|---|---|---|---|
| 0 | 5.58 | 9.54 | 44.1 |
| 1 | 5.44 | 9.51 | 43.6 |
| 10 | 5.35 | 9.59 | 42.8 |
| 20 | 5.27 | 9.51 | 43.1 |

For completeness an additional experiment on LSUN churches $64 \times 64$. Results are similar to the higher resolution case, the Blurring Diffusion Model with $\sigma_{B,\max} = 20$ achieves 2.62 FID train whereas the baseline denoising model ($\sigma_{B,\max} = 0$) achieves 2.70.

Table 7: Results on LSUN $64 \times 64$

| $\sigma_{B,\max}$ | FID train | FID eval |
|---|---|---|
| 0 | 2.70 | 44.1 |
| 20 | 2.62 | 43.1 |

