# OpenReview forum: "Blurring Diffusion Models"
_ICLR.cc/2023/Conference — ICLR 2023 poster_

### Official Review · Reviewer_HZtN · 2022-10-23

**Confidence:** 2
**Correctness:** 3
**Technical Novelty And Significance:** 2
**Empirical Novelty And Significance:** 1
**Recommendation:** 5

**Clarity, Quality, Novelty And Reproducibility:**

The connection with Ho et al 2020 is clear but not with Rissanen 2022.
The proposed insight, i.e. the frequency domain aspect on Rissanen is novel to me.
The authors gave no information whether the codes will be made available upon publication.


**Strength And Weaknesses:**

The major contribution, to the best of my understanding, lies in the analysis of IHDM drawing its connection to denoising diffusion in frequency space, as well as the proposed latent variable model. Another interesting insight is that other deterministic transformations may be also applicable for destroying the signal.
However, I’m not completely sure about the significance of the contribution, especially in comparison with Rissanen et al. 2022. Perhaps I missed something important but it would be great if the authors could highlight the technical improvement / difference.
I’m not sure if the experimental result is convincing enough either. Checking the Ho et al. 2020, I realized that their simple loss already yielded an FID of 3.17 on CIFAR10, which is exactly the same as the reported performance of the blurring diffusion model.


**Summary Of The Paper:**

This paper analyzes the connection between two diffusion models of DDPM and IHDM. Showing that the heat dissipation process is a Gaussian diffusion one in the frequency domain. Based on this insight, the authors propose the “blurring diffusion model”, which explicitly defines latent variables of the intermediate diffusion steps in the frequency space. Experiments show comparable results with the DDPM and improved ones compared with IHDM.

**Summary Of The Review:**

In general this is a piece of interesting insight but the presentation could have been better. The distinguishment to most related work is not clear to me and I wish that the authors could address my questions in the section above.

---

> ### Author Response · Authors · 2022-11-11
> **author response**
>
> We thank the reviewer for their time and address the raised questions below:
>
> 1. We have added the technical implications of our work at the end of section 2. Essentially: our derivation ensures that we have the same correctness guarantees that we have in standard diffusion, meaning that that the model p(z_s |  z_t) is able to learn q(z_s | z_t) → q(z_s | z_t, x=E[x | z_t]) as s approaches t, because now we have been able to derive the exact (non-isotropic) noise distribution of q(z_s | z_t, x). Hopefully this clears up the connection with Rissanen et al, (2022).
>
> 2. On the comparison with Ho et al., (2020): In their paper a different architecture is used that has one more level, albeit with less residual blocks. Furthermore, perhaps less obviously, they use a clipping heuristic that improves FID. In this paper, we aimed for a fair comparison and simply trained a diffusion model with the same settings without any further heuristics. Regarding the code, even though we do not plan to release the code, many great open-source implementations exist. In addition, Appendix A provides the necessary computations that are specific to diffusion with blurring, and includes precise details such as how we use small numerical values for numerical stability of division.

---

### Official Review · Reviewer_PRJZ · 2022-10-25

**Confidence:** 4
**Correctness:** 2
**Technical Novelty And Significance:** 3
**Empirical Novelty And Significance:** 3
**Recommendation:** 5

**Clarity, Quality, Novelty And Reproducibility:**

Clarity: 3/10,
Quality: 4/10,
Novelty: 5/10,
Reproducibility 6/10.

**Strength And Weaknesses:**

Strength:
(1) It makes sense for combining heat dissipation and blurring into the diffusion model.
(2) The model can generate images with lower FID.

Weaknesses:
(1) The related work part needs to be polished. The previous works are not sufficient for heat dissipation, blurring, Gaussian noise, etc.
(2) The experimental results are not stable.
  a. I can not view the generated images with high resolution.
  b. The FID is the only metric used in this paper.
  c. There is no user study.

**Summary Of The Paper:**

The paper proposes a Blurring Diffusion Model. It is a type of diffusion model based on heat dissipation, or blurring. The model obtains better performance compared with some SOTA models (e.g., Denoising Diffusion, IHDM).

**Summary Of The Review:**

The idea of the blurring diffusion model is good. And the FID values show that the proposed model has better performance than previous models. While the paper needs to be polished, especially for the experiment part.

---

> ### Author Response · Authors · 2022-11-11
> **author response**
>
> We thank the reviewer and address the points below:
> 1. On polishing the related work, we have added several more references. Please let us know if there are others that should be added.
>
> 2. On the experimental results. We have turned “interpolation” off for pdf-viewers and fixed the cifar10 png file, which means that for both figures they should now be correctly rendered. Latex uses lossless pngs that exactly match the required pixels. Unfortunately, pdf-viewers can decide themselves to still render certain images with interpolation (e.g. macOS preview does this), and there is not another way to enforce this behavior. Let us know if the issue persists and we will find another way to send the images anonymously. We have also computed and reported Inception scores for the CIFAR10 experiment in the Appendix.

---

### Official Review · Reviewer_6MNZ · 2022-10-25

**Confidence:** 4
**Correctness:** 3
**Technical Novelty And Significance:** 3
**Empirical Novelty And Significance:** 3
**Recommendation:** 6

**Clarity, Quality, Novelty And Reproducibility:**

The idea is basically to restructure Rissanen's work and combine it with denoising diffusion models. Even so, the idea is novel and interesting.

For comments on clarity and things to improve, see 'weaknesses' above.

The results appear reproducible, even if no actual code (beyond the pseudo-code in the appendix) was shared as part of the submission. Do the authors plan to release code to reproduce their experiments?

**Strength And Weaknesses:**

*Strengths:*

1. This is an interesting paper that bridges the gap between inverse heat dissipation (recently proposed by Rissanen et al.) and denoising models. The novelty lies in formulating the (de)blurring model as a Gaussian diffusion process with non-isotropic noise. This formulation is novel and complements the work by Rissanen.

2. The experiments are brief, but demonstrate the strengths of the proposed approach. The visual quality of the generated samples (as well as FID scores) is impressive---especially if compared to that given by vanilla IHDM.

3. The paper is well-structured and easy to follow.

*Weaknesses:*

4. The paper builds directly on Rissanen et al. (2022) for the deblurring part and Song's and Ho's work on denoising. The paper is written almost more as a technical note or technical comment on these existing works than an actual full paper. This, however, makes it easy to position the paper and follow the idea.

5. The paper focuses largely on technical details, and it would have benefited from expanding on discussion points (e.g., the regularisation effect briefly covered in Sec. 6 and 7). Points on the lack of 'guarantees of convergence to the data distribution' (Sec. 1) appear vague and could be either expanded, removed, or rephrased. Additional discussion on the role of the neural network architecture and its role would have been interesting.

6. Even if the presentation itself is clear, it appears slightly sloppy in parts, or written in haste. I would the authors to carefully go through the paper and fix in-text citations to read well.

**Summary Of The Paper:**

A paper bringing together aspects from diffusion (noising) and heat dissipation (blurring)  in a best-of-both-worlds fashion. Proof-of-concept results deliver impressive image quality.

**Summary Of The Review:**

This is a paper that is by no means perfect, but should be of interest to the audience of ICLR. The idea in itself is simple and extends the recent work by Rissanen, but it is a novel and impactful one.

---

> ### Author Response · Authors · 2022-11-11
> **author response**
>
> We thank the reviewer for their time and address the raised weaknesses below:
>
> 4. On positioning the paper with respect to existing works. The paper was intended to bridge the gap between the knowledge that already exists for denoising diffusion, and the new emerging field of different (correlated) diffusion processes, with emphasis on blurring. The positioning was chosen to be in direct relation to these methods, to clear up possible misconceptions about using blurring as a diffusion model.
>
> 5. We have expanded on the “lack of guarantees of convergence to the data distribution” at the end of section 2. Essentially, the point is that if p(z_s | z_t) is of the form q(z_s | z_t, x=x_prediction) (which we derive is non-isotropic) then the model can approximate q(z_s | z_t) arbitrarily accurately as s tends to t. With a different specification, such as the one in Rissanen et al., this is generally not possible. We have also added a sentence on the regularization in the first paragraph of section 4.
>
> 6. We have gone over the text to improve readability and to fix (in-line) citations. Please let us know if there are still issues that persist in this respect.

---

### Official Review · Reviewer_oaCX · 2022-10-26

**Confidence:** 3
**Correctness:** 3
**Technical Novelty And Significance:** 4
**Empirical Novelty And Significance:** 3
**Recommendation:** 8

**Clarity, Quality, Novelty And Reproducibility:**

This paper is clearly written and has both theoretical and empirical contributions.


**Strength And Weaknesses:**

Strength:
1. The authors derive an equivalence between IHDMs and standard diffusion models.
2. This is later used to significantly improve the performance of heat dissipation based models.
3. Other theoretical contributions: a) heat dissipation can be a markov process by explicitly giving one formulation, and (b) inverse heat dissipation, unlike inverse diffusion,  is not an isotropic process (in the frequency domain).

Weaknesses:
1. In pixel space, it has been shown that epsilon formulation is better. It would be a good ablation to show this holds in the frequency domain as well.
2. Page 5 last line: “ …it is convenient to express our diffusion and denoising processes in frequency space, neural networks tend to operate well on a standard pixel space”. Either there should be references supporting this or even better, an ablation study showing that neural networks perform better on pixels.
3. The significance of the two theoretical contributions namely a) inverse heat dissipation not being isotropic and b) possibility of heat dissipation being a markov process is not discussed.

**Summary Of The Paper:**

The presented work establishes a theoretical link between standard diffusion models and Inverse heat dissipation models (IHDMs). They show that IHDM formulation can be equally represented as a standard diffusion model, albeit in the frequency domain. They use this theory to apparently get the best from both approaches: (a) They are able to work with blurring which standard diffusion models are not, and (b) use epsilon parametrization which has been empirically shown to outperform the basic representation for standard diffusion models. IHDM could not use it since it was not a diffusion model.
The utility of the presented approach is demonstrated on two datasets, improving the performance of heat dissipation models significantly.

**Summary Of The Review:**

A good quality paper which theoretically links heat dissipation models with diffusion models and use that to improve the heat dissipation model. Some more work can be done on ablation studies.

---

> ### Author Response · Authors · 2022-11-11
> **author response**
>
> We thank the reviewer for their time, in the following we address the weaknesses:
>
> 1. On x-parametrization, we have run an ablation using x-parametrization on CIFAR10 for the best performing model, and achieve 23.9 FID versus 3.17 with eps-parametrization.
>
> 2. On pixel space vs frequency space, we have run an experiment on frequency space. We can see that the model is not able to learn the task correctly and samples are grey (for completeness, 260.5  FID). Also, we have rephrased the section slightly which now reads: “neural networks have been optimized to work well in standard pixel space”
>
> 3. We added the consequences of these contributions to the end of section 2. In short: it allows us to analytically derive the required noise instead of heuristically choosing it. Furthermore, this ensures the same correctness we have in standard diffusion. Meaning that the model p(z_s |  z_t) is able to learn q(z_s | z_t) → q(z_s | z_t, x=E[x | z_t]) as s approaches t.

---

> > ### Comment · Reviewer_oaCX · 2022-11-15
> > **Where to find new results (ablation outcomes)?**
> >
> > Can you please point us to the places in the main text or supplement where you report on the outcome of the discussed ablations?
> > Thanks!

---

> > > ### Author Response · Authors · 2022-11-15
> > > **location of ablations**
> > >
> > > The results had been added to Appendix B. We also included a description of the frequency experiment now.

---

### Decision · Program_Chairs · 2023-01-20

**Decision:**

Accept: poster

**Justification For Why Not Higher Score:**

The experiment results are not strong enough to prove this method will change the way people implement diffusion models.

**Justification For Why Not Lower Score:**

The experiment results support the proposed idea, which is interesting and contemporary.

**Metareview: Summary, Strengths And Weaknesses:**

The paper first shows that the inverse heat dissipation model can be considered a non-isotropic diffusion process in the spectral space. It then presents a generative model that is inspired by this realization. The generative model borrows the strength from DDPM and the idea from the inverse heat dissipation work. The presented results verify that the presented generative model works.

The paper received a mixed rating. Two reviewers consider the paper above the bar, citing the idea as interesting and the improvement over the inverse heat dissipation model major. The two other reviewers consider the paper slightly below the bar, citing issues including insufficient experiment results and unclear technical contribution. This puts the paper in the borderline category.

The AC conducted an AC-reviewer meeting. The strength and weaknesses of the paper were discussed intensively in the meeting. Based on the arguments made in the meeting, the AC is in favor of accepting the paper. First, the presented connected to the inverse heat dissipation model is interesting. Second, the presented algorithm seems to be working. While the experiment and presentation quality could be improved, the paper is interesting enough for the ICLR community.

**Note From Pc:**

if the above contains the word "oral" or "spotlight" please see: "oral" presentation means -> notable-top-5% and "spotlight" means -> notable-top-25%. As stated in our emails, we are disassociating presentation type from AC recommendations

**Summary Of Ac-Reviewer Meeting:**

3 out of 4 reviewers attended the meeting. The one, who could not make it, needs to attend a last-minute meeting.

In the AC-reviewer meeting, we asked each reviewer to comment on the number one weakness and then the number one strength. From the comments, we tried to gauge better where this paper lies. In the weakness comment round, the major weakness includes that the paper could present more results, and the paper looks like a last-minute rush. In the strength comment round, the major strength includes the presented idea being interesting enough and the presented results validating the proposed idea.

The AC took a stance and argued the paper was above the bar and asked for arguments to reject the paper. No convincing arguments against the acceptance were given.